# Identification and Characterization of Alternatively Spliced Transcript Isoforms of *IRX4* in Prostate Cancer

**DOI:** 10.3390/genes12050615

**Published:** 2021-04-21

**Authors:** Achala Fernando, Chamikara Liyanage, Afshin Moradi, Panchadsaram Janaththani, Jyotsna Batra

**Affiliations:** 1Faculty of Health, Institute of Health and Biomedical Innovation, School of Biomedical Sciences, Queensland University of Technology, Brisbane QLD 4059, Australia; achala.vitharanage@hdr.qut.edu.au (A.F.); chamikara.liyanage@hdr.qut.edu.au (C.L.); afshin.moradi@hdr.qut.edu.au (A.M.); panchadsaram.janaththani@qut.edu.au (P.J.); 2Translational Research Institute, Queensland University of Technology, Brisbane QLD 4102, Australia

**Keywords:** prostate cancer, IRX4, alternative splicing, transcript, isoforms

## Abstract

Alternative splicing (AS) is tightly regulated to maintain genomic stability in humans. However, tumor growth, metastasis and therapy resistance benefit from aberrant RNA splicing. Iroquois-class homeodomain protein 4 (IRX4) is a TALE homeobox transcription factor which has been implicated in prostate cancer (PCa) as a tumor suppressor through genome-wide association studies (GWAS) and functional follow-up studies. In the current study, we characterized 12 IRX4 transcripts in PCa cell lines, including seven novel transcripts by RT-PCR and sequencing. They demonstrate unique expression profiles between androgen-responsive and nonresponsive cell lines. These transcripts were significantly overexpressed in PCa cell lines and the cancer genome atlas program (TCGA) PCa clinical specimens, suggesting their probable involvement in PCa progression. Moreover, a PCa risk-associated SNP rs12653946 genotype GG was corelated with lower IRX4 transcript levels. Using mass spectrometry analysis, we identified two IRX4 protein isoforms (54.4 kDa, 57 kDa) comprising all the functional domains and two novel isoforms (40 kDa, 8.7 kDa) lacking functional domains. These IRX4 isoforms might induce distinct functional programming that could contribute to PCa hallmarks, thus providing novel insights into diagnostic, prognostic and therapeutic significance in PCa management.

## 1. Introduction

Alternative splicing (AS) in precursor mRNA plays a vital role in the regulation of gene expression by expanding the coding capacity of genomes. Diverse combinations of splice sites and alternative promoters in pre-mRNA are chosen to produce structurally distinct mRNA and, thus, protein isoforms that range from slightly different to having the opposite functions [1]. Different mechanistic modes of AS have been identified such as exon skipping, retention of introns, alternative 5′/3′ splice sites, alternative promoters, alternative polyadenylation sites and alterations in spliceosomes [1]. Apart from the contribution to the greater diversity of the proteome, AS leads to every hallmark of cancer progression [2]. Recent study findings predict that the vast heterogeneity of human cancers may be a result of the distinct roles of protease isoforms resulting from AS [3]. AS has been reported to modify the network of protein interactions in a disruptive, non-regulated manner, thus disrupting normal cell function via mediating cancer driven pathways [2]. Additionally, AS may induce degradation of the tumor suppressor transcripts by non-sense-mediated decay or induce mutations on the splice-sites, thus having intron-retention in tumor suppressors [2]. AS can considerably alter the coding region of the drug targets of proteins, which leads to drug and therapy resistance in many cancers [4]. Although the activities of transcription factors are extremely and coordinately regulated during embryonic growth and differentiation, AS can alter the transcriptional regulation and may switch cells from physiological to pathological transformation [5].

In 2020, prostate cancer (PCa) was the second most commonly diagnosed and the fifth leading cause of cancer-related death among men worldwide [6]. Alternative splicing plays a significant role in PCa, regulating malignant progression, aggressiveness, tumor cell lineage plasticity and therapy resistance [7,8]. Although androgen receptor (AR) splicing has been well studied in PCa, the knowledge on AS of other PCa oncogenes is scarce [9]. Continuous attempts to understand and translate the expertise in AS to PCa may accelerate the discovery of novel diagnostic and therapeutic targets to improve PCa patient care.

Iroquois-class homeodomain protein IRX4, also known as Iroquois homeobox protein 4, encoded by the gene *IRX4*, is a member of the homeobox gene family [10]. Homeobox family genes mainly act as transcription factors and are found in almost all multicellular organisms [11]. They play a crucial role in the regulation of many aspects of embryonic development including pattern formation [11]. The *homo sapiens IRX* genes consist of six members and are organized as two clusters containing three genes each: *IRX1, 2* and *4* cluster on chromosome 5 and *IRX3, 5* and *6* cluster on chromosome 16, which are separated by large intergenic regions [12]. *IRX4* is located at chromosome *5p15.3* locus and has been described as the most divergent member of the IRX family [10,13]. The *IRX* homeobox gene family is unique from other homeobox genes with an extra 3-amino acid loop extension (TALE) in their homeodomain (63 amino acids) and the 9-amino acid domain (the Iro box) [12]. The *IRX4* gene is expressed in different human organs including the developing central nervous system, breast, esophagus, skin, prostate and vagina, but is predominantly expressed in the cardiac ventricles [10]. IRX4 plays a crucial role in the regulation of the ventricular chamber-specific gene expression by triggering the ventricular myosin heavy chain-1 (VMHC1) gene and suppressing the atrial myosin heavy chain-1 (AMHC1) gene [14]. IRX4 supports the maintenance of cardiac contractile function and has a protective role in cardiomyopathy, cardiac hypertrophy and congenital heart disease [15,16]. In addition, *IRX4* expression was identified in the retina as a crucial regulator of *Slit1* expression [17]. Far beyond its physiological role, the differential expression and oncogenic role of IRX4 has been suggested in recent studies. High levels of IRX4 in breast cancer plasma samples have been reported, which suggest the potential of IRX4 as a biomarker for breast cancer [18]. IRX4 expression was found to be downregulated in the mesenchymal cell population compared to epithelial cells in breast cancer [19]. Overexpression of IRX4 drives cell proliferation in non-small cell lung cancer (NSCLC) and is directly associated with the overall survival of patients [20]. The *IRX4* promoter region was found to be frequently hypermethylated in pancreatic cancer, which provides an advantage for cell growth in pancreatic cancer [21]. *IRX4* has been identified recently as a potential candidate gene in PCa after genome-wide association studies (GWAS) discovered the *5p15* locus to be associated with PCa risk. Many studies carried out by Batra et al. [22], Wang et al. [23], Lindstrom et al. [24] and Qi et al. [25] have identified the association of SNP rs12653946 at *5p15*, a cis-eQTL of *IRX4,* with PCa risk in multiethnic populations. IRX4 has been described as a tumor suppressor in PCa with the interaction of vitamin D receptor [26]. The differential roles of IRXs in a tumor microenvironment have been reported [27,28,29], possibly suggesting their tissue-specific role and/or their splicing, which affect the protein level, thereby modifying the functional capacity of normal cells to prompt and withstand multiple mechanisms related to tumor progression.

*IRX4* is a human multi-exon transcription factor, which has been reported as alternatively spliced and highly expressed in PCa [26]. To date, five transcripts of *IRX4* have been identified, including four predicted alternative promoters [26,30]. *IRX4* has mainly two types of transcripts that differ from the presence of an additional fourth exon, further divided into five transcripts differing only from the 5′UTR region, and the expression of these transcripts have been reported in PCa [26]. The total of five transcripts are predicted to translate into two IRX4 protein isoforms that only differ from the presence of an additional 27 amino acids that is encoded by an extra fourth exon (UniProt database),functional role of which is still unknown. In this article, we present the diversity of *IRX4* transcripts and IRX4 protein expression in PCa cell lines and clinical samples, as IRX4 protein isoforms may have differential roles in PCa progression. This study enables a broad understanding of the regulation and mechanisms in prostate tumorigenesis, which is very limited at present.

## 2. Materials and Methods

### 2.1. Cell Culture

A panel of PCa cell lines (LNCaP, VCaP, DuCaP, C42B, PC3, DU145, RWPE2, 22RV1) and benign prostate (BPH1, RWPE1) cell lines were purchased from the American Type Culture Collection (ATCC, Manassas, VA, USA). RWPE-1 and RWPE-2 cell lines were grown in keratinocyte serum-free medium supplemented with 5 ng/mL recombinant human epidermal growth factor (EGF) and 50 μg/mL bovine pituitary extract (Gibco™, Invitrogen, Carlsbad, CA, USA), whereas all other cell lines were grown in RPMI1640 (1X) with no phenol red (Life Technologies, Grand Island, NY, USA) supplemented with either 5% or 10% fetal bovine serum (FBS, Life Technologies, Thornton, NSW, Australia). The cell lines were authenticated by short tandem repeat (STR) profiling and tested negative for mycoplasma. The cells were maintained at 37 °C in a 5% CO_2_ humidified incubator.

### 2.2. RNA Isolation and cDNA Synthesis

Total RNA was extracted from PCa cells using the Isolate II RNA Mini Kit (Bioline, London, UK) according to standard protocol. RNA concentration and purity were measured using NanoDropTM1000 (Thermo Scientific, BiolaB, Scoresby, VIC, Australia). A total of 1 μg of RNA was reverse transcribed to cDNA using SensiFast^TM^ cDNA synthesis kit (Bioline, GmbH, Luckenwalde, Germany). The cDNA was diluted to 100 μL before using it as a template for PCR reaction.

### 2.3. Reverse Transcription-Polymerase Chain Reaction (RT-PCR)

The primers for RT-PCR and qRT-PCR were designed using NCBI tool Primer BLAST–NCBI–NIH software. Several primer sets were designed to recognize the boundary between 2 exons (exon-exon spanning region) to specifically identify the expression of the *IRX4* transcripts. All the primer sequences are given in Appendix A. RT-PCR was performed with a reaction comprising 1X PCR buffer, 1.5 mM MgCl_2_, 0.2 mM dNTPs and 0.2 μM of each of forward and reverse primers (Sigma Aldrich, Castle Hill, NSW, Australia), 1 U Platinum™ Taq DNA Polymerase (Invitrogen, Carlsbad, CA, USA) and 1 μL of th cDNA template. PCR reaction was performed on a Mastercycler^®^ nexus machine (Eppendorf, North Ryde, NSW, Australia). The samples mixed with loading dye (NEB #B7024S) were loaded on to 0.7–2% agarose gels (Bioline, Alexandria, NSW, Australia) prepared in Tris-borate-EDTA (TBE) buffer (89 mM Tris base, 89 mM Borate, 2 mM EDTA) containing 0.5 μg/mL ethidium bromide (Invitrogen, Carlsbad, CA, USA). Approximately 0.5 μg of 1 kb ladder (New England Biolabs, Ipswich, MA, USA) was loaded to compare the size of the DNA products. Images were captured by the gel documentation system QUANTUM ST5 (Fisher Biotec, Wembley, WA, Australia).

### 2.4. Relative Quantification by Real-Time Quantitative RT-PCR (qRT-PCR)

Quantitative RT-PCR was performed using the ViiA7 Real-Time PCR system (Applied Biosystems, Foster City, CA, USA). Each reaction contained 1X final concentration of SYBR Green PCR Master Mix 2X (Applied Biosystems, Foster City, CA, USA), 50 nM forward and reverse primer, 2.0 μL of diluted cDNA (1:5) and nuclease-free water at a final volume of 8 μL. The cycling parameters were 95 °C for 10 min, 40 cycles of 95 °C for 15 s and 60 °C for 1 min followed by a dissociation step. All the CT values were normalized to the expression of housekeeping gene *RPL32* (ΔCT) [31]. Relative expression compared to control was performed by the comparative CT (ΔΔCT) method.

### 2.5. Androgen Deprivation Assay

LNCaP, VCaP and DuCaP cells were seeded in RPMI1640 media (Life Technologies, Grand Island, NY, USA) supplemented with 5% FBS and incubated at 37 °C for 3 days. The medium was then replaced with an androgen-depleted culture medium (RPMI1640) containing 5% charcoal-stripped serum (CSS, Sigma-Aldrich, Castle Hill, Australia). After 48 h, the cells in CSS were supplemented with 10 nM dihydrotestosterone (DHT), 10 nM DHT + 10 μM anti-androgens (bicalutamide or enzalutamide) and ethanol (EtOH) control and incubated at 37 °C for 48 h.

### 2.6. In Silico Analysis of IRX4 Transcripts and Isoforms

The UCSC genome browser (https://genome.ucsc.edu/accessed on 24 November 2019) [30], the National Center for Biotechnology Information (NCBI) (https://www.ncbi.nlm.nih.gov/accessed on 18 October 2019) [32], GTEx portal (https://gtexportal.org/home/accessed on 10 January 2020), The Human Protein Atlas database (https://www.proteinatlas.org/accessed on 25 January 2020) and UniProt database (https://www.uniprot.org/accessed on 12 July 2020) were used to obtain *in silico* data for the expression of *IRX4* transcripts and protein isoforms.

### 2.7. RNA-seq and Genotype Data for cis-eQTL Analysis

The RNA-seq data (bam format) of 483 PCa patients and 49 matched controls of The Cancer Genome Atlas Program (TCGA) were used in the study [33]. RASflow was used for RNA-seq analysis [34]. HISAT2, a fast and sensitive alignment program, was selected for alignment to the transcriptome [35]; feature-count was utilized in the quantification, and Deseq2 was used in the normalization of data. Cleaned genotypes data of PCa risk SNP rs10866528 were used for cis-eQTL analysis.

### 2.8. Reprocessing PRIDE PCa Cell Line LC-MS/MS Data

LC-MS/MS raw data files of the LNCaP cell nuclear extractions were retrieved from the Proteomics Identification Database (PRIDE) database, belonging to the project ID: PXD003262 [36]. All files were converted to MASCOT generic format (MGF) peak files using MSConvertGUI (Version 3) [37]. Next, all peak files were searched in SearchGUI (Version 3.3.17) and PeptideShaker (Version 1.16.43) against a FASTA database comprising novel IRX4 peptide sequences merged with UniProt human reference database and contaminant proteins [38,39]. X! Tandem search algorithms were implemented using the following search parameters: precursor mass error: 10 p.p.m., fragment mass error: 0.05 Da, fixed modification: carbamidomethylation of cysteine, variable modification: oxidation of methionine defined as variable modification. Minimal peptide length was set to six amino acids. The 1%-fold discover rate (FDR) was set to identify specific tryptic peptides representing each IRX4 protein isoform.

### 2.9. DNA Sequencing

The PCR products were purified by Wizard^®^ SV Gel and PCR Clean-Up System (Promega, Madison, USA) and sequenced by AGRF (Gehrmann Laboratories, Research Rd, University of Queensland, Brisbane, Australia). A total of 11 µL of the purified PCR products were sent with 1 µL primers (10 µM) in standard 1.5 mL Eppendorf tubes, and the results were obtained via the AGRF online website. For the identification of amplified DNA fragments, DNA sequences were aligned against the NCBI database [32] and UCSC genome browser [40].

### 2.10. LC-MS/MS Analysis of PCa Cells

Cell pellets were obtained and lysed using sodium deoxycholate (SDC) buffer (1% SDC in 1M Tris pH 8.0). The samples were then sonicated in an ultrasonic bath (Thermo Scientific™, Waltham, MA, USA) for 15 min (at 4 °C, 100% Power) to denature proteins and shear DNA. The concentration of proteins was calculated using a bicinchoninic acid assay (BCA) with Pierce™ Bovine Serum Albumin (BSA) Standards (Thermo Scientific™). A total of 10 μg of the protein extract was denatured at 95˚C for 5 min using a thermomixer (Eppendorf ThermoMixer^®^ F1.5, Hamburg, Germany). Denatured protein samples were reduced by 10 mM Tris (2-carboxyethyl) phosphine (TCEP) (Sigma-Aldrich, Castle Hill, NSW, Australia), alkylated by 40 mM 2-chloroacetamide (2CAA) (Sigma-Aldrich) and incubated for 30 min in the dark at room temperature. Samples were then digested overnight at 37 °C by adding trypsin (Sigma-Aldrich) at a 1:50 enzyme-protein ratio. Peptides were desalted using Pierce™ C18 Spin Tips (Thermofisher, Waltham, MA, USA), washed in 0.1% TFA, and peptides were dissolved in 80% acetonitrile (ACN) (HPLC grade, Sigma-Aldrich) elution buffer. Solvents were evaporated in a SpeedVac centrifuge (Savant Speed Vac, SPD121P-230, Thermo Electron Corporation, Milford, MA, USA) at 35˚C and re-suspended using iRT calibration mix including 2% ACN ans 0.1% TFA (Biognosys AG, Schlieren, Switzerland). Samples were prepared in 3 biological replicates and analyzed by a sequential window acquisition of all theoretical mass spectra (SWATH-MS) approach, following our previously published protocol [41].

### 2.11. LC-MS/MS Data Analysis

Data-dependent acquisitions were imported into ProteinPilotTM software (Version 5.0.1, AB SCIEX) and searched using the Paragon™ algorithm against the FASTA database consisting of novel IRX4 peptide sequences merged with the UniProt human reference database and contaminant proteins. The following search parameters were used: Sample type: Identification; Cys Alkylation: Iodoacetamide; Digestion: Trypsin; Instrument: TripleTOF 5600+; Species: None; Search effort: Thorough ID; Results Quality: Detected protein threshold [Unused ProtScore (Conf)] ≥ 0.05 with FDR. Generated ion library was imported into PeakView^®^ SWATH micro app (Version 2.1, AB SCIEX) and saved in text format and cleaned using the iSwathX tool (Version 2.0). The curated ion library was imported into Skyline software (Version 1.1) [42]. The following peptide and transition settings were followed: Enzyme: Trypsin [KRǀP]; Max missed cleavages: 1; Min length: 6; Max length: 35, Precursor charges: 2+,3+,4+; Ion Charges: 1+,2+,3+; Ion types:y/b; Ion match tolerance: 0.5. Data-independent acquisitions were imported, and peptide quantification was performed using MSstats R-based statistical tool (Version 2.0) [43]. Peptides only specific to each IRX4 protein isoforms were used for the quantification, and normalized peptide intensities were used to calculate the relative fold expression.

### 2.12. Statistical Analysis

All statistical data were analyzed by GraphPad Prism 9.0.0 (121). The comparison was analyzed by paired t-test (two groups) and Kruskal–Wallis test with Dunn’s multiple comparisons (more than two groups). The results were considered statistically significant if * *p* < 0.05 at a 95% confidence interval. All the experiments were performed in 3 biological replicates.

## 3. Results

### 3.1. In silico identification and characterisation of human IRX4 transcripts

The human *IRX4* gene at *5p15.33* locus comprises six coding exons, seven introns and 3′UTR and 5′UTR regions, which spans from 1,877,413 to 1,887,236 bp (hg38) [32,40]. Although not fully characterized, the IRX4 protein consists of a homeodomain or DNA binding domain, transactivation domain and Iro box (Figure 1a). Five *IRX4* transcripts have been reported for *IRX4* to date in NCBI [32]. The transcript 1(NM_001278632.1) and the 3(NM_001278634.2) and 5(NM_016358.3) transcripts differ from the 2(NM_001278633.1) and 4(NM_001278635.2) transcripts since the latter transcripts consist of an additional fourth exon (78 bp). *IRX4* transcripts 1, 3 and 5 are similar in their coding regions but differ from each other with their diverse 5′UTR region, and also the 2 and 4 transcripts are similar in their coding regions and contrast with each other with their different 5′UTR region (Figure 1b) [30]. According to the alternative splicing graph of *IRX4* by the Swiss Institute of Bioinformatics (Figure 1c), there is clear evidence that *IRX4* has been alternatively spliced and has a probability to encode different transcripts (Figure 1c) [30].

Four alternative promoters were predicted to be localized upstream of the 5′UTR of *IRX4* by the UCSC genome browser: the alternative promoter 1 (chr5:1887187–1887336), the alternative promoter 2 (chr5:1887169–1887318), the alternative promoter 3 (chr5:1882876–1883025) and the alternative promoter 4 (chr5:1881044–1881193), as shown in the Figure 1b [30]. The predicted 1 and 2 alternative promoters and the presence of an EST (AI246240) in the 5′UTR region facilitated the identification of novel 5′UTR exons (exon 1a and 1b) (Figure 1b). Nguyen et al. suggests the presence of a diverse 5′UTR region of *IRX4*, later confirmed by Northern blot analysis and RACE [26]. Further, few additional transcripts of *IRX4* have been predicted by the Swiss Institute of Bioinformatics gene predictions based on the mRNA and EST expression [30]. A novel transcript (PV1) (HTR011738.5.69.3) with a putative novel exon (exon 3a, 89 bp) localized between the second and third exons has been predicted (Figure 1b). The presence of this putative exon is also reported for *IRX4* in an EST dataset (BY799479) and the alternative splicing prediction plot of *IRX4* by Swiss Institute of Bioinformatics (Figure 1c). The expression of this exon has been reported in the skin with no sun exposure and esophagus mucosa according to the GTEx portal data. Moreover, the presence of an alternative promoter region (alternative promoter 4) has been predicted in this region, which further confirms a new starting point for this transcript (Figure 1b). Another short transcript (PV2) (HTR011738.5.69.2) starting from a novel open reading frame (ORF) from exon 5 of the *IRX4* has also been reported (Figure 1b) [30]. The 5′UTR region of this transcript consists of four exons, slightly differing from the other transcript exons by retention of part of the intron 6 (I6) and longer first UTR region. The 3′UTR end of both predicted transcripts is similar to that of other transcripts (Figure 1b). The *IRX4* 1-4 transcripts may deploy the alternative promoter 1 and 2; the transcript 5 utilizes the alternative promoter 3, while PV1 makes use of alternative promoter 4. The existence of alternative promoters reflects the possibility of multiple transcripts for the *IRX4* gene that differ in their size and regulate separately at their transcription level.

Expression of most of the *IRX4* exons f was observed across several tissue types such as skin, prostate, salivary glands, esophagus, vagina and heart, but the last exon (exon 6) was abundantly expressed in most of the tissues studied (GTEx portal data). IRX4 proteins were mainly localized in the nucleus and vesicles. Enhanced expression of IRX4 has been detected in some human cell lines, such as brain cancer (BEWO), skin immortalized (HaCaT), lung immortalized (HBEC3-KT) and breast cancer (MCF7) cell lines (the Human Protein Atlas database).

Even though IRX4 has been identified for its transcriptional role in the human heart and tumor suppressor role in PCa, the IRX4 proteins are so far not fully structurally and functionally characterized for their interactions and the functional domains. Three domains have been predicted to date: the homeodomain (DNA binding domain) (63aa), the putative transactivation domain (18aa) and Iro box (9aa) for the IRX4 protein (Figure 1a). The homeodomain and the transactivation domain of IRX4 are encoded from part of the fifth exon of the transcript, and the iro box region is encoded from part of the sixth exon [14]. The IRX4 protein isoform 1 (Uniprot ID- P78413-1 length: 519 aa; mass: 54.4 kDa) is encoded from the transcripts 1, 3 and 5 and differs only at 26 amino acids with IRX4 protein isoform 2 (Uniprot ID- P78413-2 length: 545 aa; mass: 57.0 kDa), which is encoded from transcripts 2 and 4. The addition of 26 amino acids in isoform 2 may result in a structural change that may affect the function. A panel of PCa cell lines representing the benign and metastatic nature has been used in the study. All the features corresponding to each PCa cell line have been characterized in Figure 1d.

### 3.2. Identification of Human IRX4 Transcripts in PCa Cell Lines

Firstly, we tried to identify the expression of five known *IRX4* transcripts in a panel of PCa cell lines. Two main bands (369 bp and 291 bp) on the agarose gel were detected in RT-PCR with the primer set 1, designed to amplify the region from complementary to exon 2 to 5, confirming the presence of two transcripts, where one transcript contains a fourth exon and the other does not (Figure 2a,f). The sequencing analysis of the resulting bands confirmed the presence of at least two *IRX4* transcripts with and without exon 4 in PCa cell lines The upper band may correspond to either of the transcripts 2 and 4 (Figure 2a). The lower band may correspond to either of transcripts 1, 3 and 5.

Next, we investigated the expression of the predicted novel transcript 1 (PV1) (HTR011738.5.69.3, transcript 6) in C42B and LNCaP cells. The presence of this transcript was identified by RT-PCR after designing a specific primer set that targets the novel exon (exon 3a) and fifth exon (primer set 2). Unexpectedly, two bands higher (about 451 bp and 529 bp) than the expected size (281 bp) were detected (Figure 2b). The Sanger sequencing analysis of the purified PCR product identified the intron retention (I4) between the novel exon (exon3a) and the third exon. The Sanger sequencing analysis of the purified product of the upper band confirmed the existence of the additional fourth exon suggestive of an AS event of transcript 6 (Figure 2b,f). For subsequent identification, the transcript that corresponds to the upper band was termed transcript 7 (Figure 3).

Next, we were interested to know whether the novel exon (exon 3a) is the first exon of transcripts 6 and 7. We designed a specific primer set complementary to the second exon and the novel intron region (I4) (primer set 3) and checked the expression in C42B and LNCaP cells. Interestingly, we found the presence of a band around 176 bp (Figure 2c) in PCR, but below the expected size (265 bp). Sequencing analysis confirmed a new *IRX4* transcript (which designated as variant 8) corresponding to this band (Figure 2c, f). This transcript lacks the novel 3a exon but has the intron 4 (I4) region; the intron 4 and the third exon appear as a single exon, which we called exon 3b. We saw that this transcript has an AS event with the fourth exon and appears as a new transcript with the additional fourth exon, which we termed as transcript 9 (Figure 3).

Interestingly, two new transcripts were identified with primers in the second and sixth exon in DuCaP and C42B cells. Two unexpected lower bands (386 bp and 464 bp) were detected with the primer set 4, in addition to the two expected bands (544 bp and 622 bp, Figure 2d). The sequencing results confirmed that the lower bands lack part of the exon 5 (exon 5a) of *IRX4* (Figure 2d,f). The two lower bands differ from each other with the additional fourth exon. The transcript that lacks the part of exon 5 and the fourth exon is termed transcript 10, and the transcript that lacks part of the exon 5 but retains the fourth exon is termed transcript 11 (Figure 3).

We were then able to confirm the presence of the second predicted transcript (PV2) by the Swiss Institute of Bioinformatics. The transcript has an extended fifth exon (exon 5b) in its 5′UTR region; the forward primer was designed specifically for this region to exclude the expression of other transcripts (primer set 5). The expression was detected in this transcript at 396 bp as expected in few PCa cell lines (Figure 2e). This short transcript, which we denoted as transcript 12, was predicted to have a coding region starting from the fifth exon, and all other starting exons have been predicted as 5′UTR regions (Figure 3).

The relative *IRX4* transcripts 1–5 expression levels were measured by qRT-PCR with three different primer sets, corresponding to all transcripts (primer set-12), transcripts 1, 3 and 5 (primer set-13) and transcripts 2 and 4 (primer set-14) (Figure 2g). The castration-resistant PCa cell line, C42B, had the highest expression of *IRX4* (more than 20-fold) according to the primers capturing all *IRX4* transcripts (Figure 2g) compared to the expression in BPH1. The RWPE-1 and RWPE-2 prostate epithelial cell lines expressed a significantly higher expression of all *IRX4* transcripts compared to BPH1. However, a poor correlation between the intensity of gel bands (Figure 2a) and qPCR graphs (Figure 2g) was detected due to different primer efficiencies used in PCRs. Compared to androgen-responsive cell lines LNCaP, DuCaP and VCaP, androgen-nonresponsive PCa cell lines PC3, 22RV1 and DU145 had a minimal expression of *IRX4*. Interestingly, overall *IRX4* expression was significantly higher in the castration-resistant C42B cell line (a derivative of LNCaP) compared to LNCaP, indicating a role of IRX4 in castration-resistant PCa progression.

The lower band that corresponds to transcripts 1, 3 and 5 expression was also prominently detected in C42B cells (more than 25-fold) compared to BPH1, but there was a comparatively low expression in 22RV1 and PC3 cell lines (Figure 2g) consistent with the RT-PCR results. *IRX4* transcripts 2 and 4 were prominently detected in RWPE1 and 2, but were weakly detected in 22RV1, DU145 and PC3 cells, similar to the RT-PCR upper band of the gel (Figure 2g). Overall, the results of the expression of *IRX4* transcripts showed a higher disparity between PCa cell lines in relation to their androgen responsiveness, which may be suggestive of differential regulation of *IRX4* transcripts by androgen in PCa cells. Our results indicate that human *IRX4* is extremely alternatively spliced, and many *IRX4* transcripts expressions were able to be identified in most of the PCa cell lines. The expression of *IRX4* transcripts still have not been completely annotated in gene databases. All the transcripts identified for *IRX4* in PCa cells are summarized in Figure 3. The sequence alignment for transcripts 6, 8, 10 and 12 is shown in the Supplementary Data (Appendix A) concerning their altered regions.

### 3.3. Characterisation of Human Novel IRX4 Transcripts in a Panel of PCa Cell Lines

Differential expression of *IRX4* transcripts 1–5 expression was detected in a panel of PCa cell lines (Figure 2a). The RT-PCR primers (grey) were designed specifically to target one transcript at one time excluding all the other transcripts (Figure 4a). However, the expression of transcripts 6 and 7 (primer set 15) and transcripts 8 and 9 (primer set 16) was not distinguishable designing specific qRT-PCR primers (green), but transcripts 10, 11 and 12 expression was individually measured with specific primer sets (primer set 17, 18 and 19) to obtain a relative quantification (Figure 4b). All the bands obtained were confirmed for their sequence with Sanger sequencing.

The highest expression of transcripts 6 and 7 was detected in DuCaP cells (~15 fold) compared to BPH1, consistent with the RT-PCR results (Figure 4b). However, C42B, RWPE1 and RWPE2 cells showed a significantly higher expression of transcripts 6 and 7 compared to BPH1 (Figure 4). The highest expression of transcripts 8, 9, 10, 11 and 12 was detected in C42B cells, and it reached statistical significance compared to BPH1, in line with the intensity of gel bands suggestive of a role of IRX4 in metastatic progression of PCa. (Figure 4b). The identified novel transcripts showed a minimal or no expression in 22RV1, PC3 and DU145 cell lines (Figure 4b), except for transcript 12 which has a comparable higher expression in DU145 cells. The PC3 cell line has a minimal expression of all identified novel transcripts. The results showed a high disparity of expression of *IRX4* novel transcripts between PCa cell lines. The results suggest that the identified novel transcripts significantly contribute to the overall overexpression of *IRX4* in PCa and may have a role in PCa progression.

### 3.4. Androgen Regulation of IRX4 Transcripts in PCa Cell Lines

Relative expression of *IRX4* transcripts in PCa cell lines with comparatively higher expression in androgen-responsive cell lines and low expression in androgen-nonresponsive cell lines may indicate that the expression of *IRX4* transcripts could be regulated by androgens and may have a role in therapy resistance. Thus, the expression of each *IRX4* transcript was determined in androgen-responsive cell lines (LNCaP, VCaP and DuCaP) with androgen (DHT) and antiandrogen (Bicalutamide and Enzalutamide) treatment. *KLK3* (PSA) expression was used as a positive control to validate both the treatments. *KLK3* was overexpressed with DHT treatment in all three cell lines, VCaP ~500-fold, DuCaP >10-fold and LNCaP ~40-fold and downregulated with antiandrogen treatment (Figure 5) compared to the ethanol (EtOH) control. Almost all *IRX4* transcripts expressions were upregulated with DHT treatment in VCaP and DuCaP cells compared to the EtOH control, and the upregulation of transcript 10 and 11 was prominent among other transcripts. Transcript 10 is overexpressed more than 300-fold and transcript 11 by more than 200-fold compared to EtOH with DHT in VCaP cells, and by 15-fold and 12-fold compared to EtOH in DuCaP cells, respectively, consistent with the treatment effect. Interestingly this expression pattern is not observed in LNCaP cells even with 40-fold *KLK3* overexpression (comparatively higher as compared to DuCaP) (Figure 5). The bicalutamide and enzalutamide treatment effectively reduced the androgen mediated expression of all identified *IRX4* transcripts in VCaP and DuCaP cells except for transcripts 6 amd 7 in DuCaP compared to DHT. This suggests that the antiandrogen treatment is not effective to reduce the expression transcripts 6 and 7, which show the highest expression in DuCaP cells among other cell lines. Although the expression of *KLK3* was reduced in LNCaP cells with antiandrogen treatment, a significant upregulation or downregulation of *IRX4* transcripts was not observed in LNCaP cells (Figure 5).

### 3.5. Identification and Characterisation of IRX4 Protein Isoforms by Mass Spectrometry

All the presented known and novel *IRX4* transcripts were characterized by their ORF for having the potential to encode novel protein isoforms of IRX4 using the ExPASy translate tool. The two protein isoforms, IRX4 isoform 1 and 2 encoded from transcripts 1 to 5, have been reported in the UniProt database (P78413-1 and P78413-2), and the expression has been validated in the human heart. The ORF coding site ranges from the first exon to exon 6 in *IRX4* transcripts 1 to 5 and encodes full-length proteins (54.4 kDa and 57 kDa). For most of the newly identified transcripts, the coding frame shifts from the main transcripts, and novel stop codons are incorporated (Table 1). Therefore, new transcripts produce truncated proteins. There is more than one coding frame for a few transcripts such as transcripts 6, 7, 8 and 9. The proteins generated from those novel transcripts differ from the main two protein isoforms and may lack the essential domains for their functional activity. Splice deletion of exon 1 and 2 and insertion of intron 4 (I4) produces transcripts 6 and 7 and thus has the capability to produce truncated proteins with two probable coding frames (ORFs), exon 3a–exon 5 and exon 5–exon 6 (Table 1). Moreover, transcripts 8 and 9 have intron retention between exons 3 and 3a, excluding exon 3a, and change the whole coding frame to encode a short protein isoform (11 kDa) due to the insertion of a new stop codon in their intron region (I4). The predicted second frame for these two transcripts is similar to the frame of exon 5–exon 6. Among the 12 *IRX4* SVs, two transcripts had a deletion of exon 5 (transcript 10 and 11), thus inserting a new stop codon at exon 6 which is predicted to encode two protein isoforms (21 kDa and 23.6 kDa), respectively. Transcript 12 presents the coding frame from exon 5 to exon 6 that predicted the encoding of a novel protein isoform (40 kDa). The transcripts 6, 7, 8, 9 and 12 also have the potential to encode this protein isoform. Then, we tried to identify the exact coding frame for each *IRX4* transcript by looking at their protein expression level. The details of the ExPASy tool analysis for probable protein coding frames of the identified *IRX4* transcripts and the probability to obtain their predicted domains are mentioned in Table 1.

Due to the lack of availability of IRX4 protein isoform-specific antibodies, we analyzed the predicted protein coding frames of all identified *IRX4* transcripts with the available proteomic identificationdatabase of PCa cell lines (PRIDE) with peptide shaker and seachGUI software. We could detect the MS/MS fragmentation ion spectrums of peptides that correspond to the coding frames of four IRX4 protein isoforms in LNCaP cells (Appendix A). They are the following: the first isoform (54.4 kDa) encoded by transcripts 1, 3 and 5, the second isoform (57 kDa) encoded by transcripts 2 and 4, the third isoform (40 kDa) encoded by transcripts 12, 8 or 9 and the fourth isoform (8.7 kDa) encoded transcripts 6 and 7. A peptide sequence that is specific to isoform 2 and a specific peptide sequence that is specific to isoform 4 were detected in the mass spectrometry data individually (Appendix A). We also detected the common peptides related to isoforms 1 and 3 but cannot exactly differentiate these isoforms since they share a similar sequence identity with isoform 1. Unfortunately, we did not see the expression of peptides generated from the coding frames of transcript 10 (21 kDa) and 11 (23.6 kDa) and the short coding frame of transcripts 8 and 9 (11 kDa). Thus, the fate of these short proteins is still unknown, which may be suggestive of nonsense-mediated decay of these transcripts or might be temporal. The identified peptides in the IRX4 protein isoforms and the MS/MS fragmentation ion spectrum files of the peptides are presented in Appendix A.

Following the identification of the IRX4 protein isoforms in the published PRIDE data, we intended to quantify the isoform-specific peptide expression in PCa cell lines and BPH1 using the SWATH-MS/MS approach. Specific peptides used to determine isoform-specific expression in PCa cell lines were mentioned in Figure 6. Expression of protein isoform 3 was excluded from the analysis since it does not contain a specific peptide to differentiate from other isoforms. According to the analysis, the expression of IRX4 protein isoform 1 showed significantly higher expression in the VCaP cell line (~10 fold) and LNCaP cell line (~6 fold) compared to the expression in BPH1 (Figure 6a). Although not statistically significant, we observed a moderately higher expression of protein isoform 2 in LNCaP and DuCaP cell lines (Figure 6b). The expression of IRX4 protein isoform 4 showed significantly higher expression in the LNCaP cell line (~2 fold) compared to the expression in BPH1 (Figure 6c). In summary, IRX4 protein isoforms showed a marked expression in androgen-responsive cell lines (LNCaP, DuCaP and VCaP) compared to androgen-nonresponsive cell lines (22RV1 and DU145). However, all three IRX4 protein isoforms were found highly expressed in the castration resistant C42B cell line compared to other androgen-nonresponsive cell lines used for the analysis.

### 3.6. Validation of Expression of IRX4 Transcripts in PCa Patients

Since the expression of individual *IRX4* transcripts has not been determined in patients with PCa earlier, the expression of the identified *IRX4* transcripts was analysed in 49 PCa patient sample tumors (*N* = 49) and their matched non-malignant tissues (*N* = 49) from the TCGA database. We observed significant overexpression of *IRX4* transcripts 3, 5 (protein isoform 1) and 6 (protein isoform 4) in PCa tissues compared with their non-malignant tissues (Figure 7). However, the expression of *IRX4* transcripts 2 and 4 (protein isoform 2) in PCa tumor samples did not reach statistical significance compared to their normal tissues. The number of samples that express *IRX4* transcript 1 was quite low in the TCGA database and was thus excluded from the study. Since the identified novel transcripts are still not annotated with precise IDs, we could not extract the data related to the expression levels from the RNAseq data of the TCGA patient samples.

### 3.7. Association between PCa Risk SNP rs12653946 and Expression Levels of IRX4 Transcripts

The numerous prostate cancer-associated SNPs have been identified by GWAS, and the regulation effect of these SNPs on the expression of the nearby genes is investigated by expression quantitative trait loci (eQTLs) methods [44]. Those SNPs that can influence the expression of nearby genes are called cis-eQTL. Xu *et al*. identified fifty-nine SNPs from 39 distinct PCa risk loci, among them rs12653946 SNP was found to be a cis-eQTL which has the strongest association with *IRX4* [44]. The risk-associated genotype (GG) was associated with lower IRX4 levels in PCa [44]. In view of this, we were interested to see whether the expression pattern of the *IRX4* transcript varies with PCa patients’ genotypes. rs10866528 is a tag SNP for PCa-risk SNP rs12653946, which is in linkage disequilibrium (LD), and it was later found that this SNP itself can act as a PCa-risk SNP [45]. We extracted the *IRX4* transcripts expression data of483 patients from the TCGA database and correlated them with the patients’ SNP rs10866528 genotype. According to the analyzed expression of *IRX4* transcripts 2, 4, 5, and 6. we found that the GG PCa risk-associated genotype is associated with lower levels of all the analyzed *IRX4* transcripts (Figure 8). Although we did not see a significant lower expression of *IRX4* transcript 3 with the genotype GG, the sample number that showed the expression of transcript 3 was quite low.

## 4. Discussion

Novel research insights have proved that protein isoforms are sophisticated expression-based biomarkers in cancer prediction and progression [46,47,48,49,50,51]. Systematic sequencing of the human genome and transcriptome has revealed that more than 90% of genes express multiple mRNAs via AS events, suggesting a major impact on the functional diversity of proteins [52]. However, cancer cells frequently exhibit abnormalities in RNA splicing to survive, grow and progress to therapeutic resistance [53]. Many studies have highlighted that alternative RNA splicing is a common intrinsic mechanism leading to therapy and drug resistance in PCa [8,9,54]. For example, constitutively active ARtranscript 7 (*AR-V7*) in prostate tumor cells confers a primary or an acquired resistance to androgen deprivation therapy [55]. In addition to *AR*, several other genes undergoing AS, such as *FGFR, VEGF, Bcl-x, SH3GLB1* and *CCDN1*, were found to be associated with PCa development and progression [8]. Homeodomain transcription factors are shown to frequently be alternatively spliced [56]. For example, the homeodomain gene *HNF1B*, which encodes for three protein isoforms A, B and C, has been shown to have different functions with respect to the transcripts, as HNF1B A and B protein isoforms act as transcriptional activators, while HNF1B C protein isoform that lacks the transactivation domain functions as a transcription repressor [56,57,58].

Nguyen et al. have identified four novel transcripts of *IRX4* in PCa cells [26]. Although for these four *IRX4* transcripts, the sequences of exons 1 to 6 were highly conserved, the sequences of their upstream exons, which encode the 5′UTRs, are diverse in sequence and length [26]. We now reveal considerably greater diversity in human *IRX4* transcripts than previously realized. In the present study, we identified by RT-PCR analysis 12 *IRX4* transcripts, including seven novel transcripts in PCa cells. This diversity of the *IRX4* gene mainly arises from alternative promoter usage, intron retention, exon skipping and alternative 3′ and 5′ splice site usage. The 5′UTRs of mRNAs play important roles in the posttranscriptional regulation of gene expression [59]. Nguyen et al. have identified additional exons of the *IRX4* 5′UTR region by Northern blot analysis of *IRX4* transcripts [26]. The identified novel *IRX4* transcripts are yet to be characterized for their UTR regions. This complex isoform diversity of *IRX4* gene may not be constrained to PCa but can deviate to other cancers with significant expression of IRX4; thus, it is worth exploring the pathological and/or physiological impact of them.

Androgens and AR play a critical role in PCa pathogenesis. Androgen deprivation therapy (ADT) has been the mainstay of management for advanced PCa. Despite the initial strong responses to androgen deprivation therapy, the majority of patients with advanced PCa relapse with fatal castration-resistant PCa (CRPC) [60]. According to our data, *IRX4* and its isoforms are differentially expressed depending on the cell lines’ androgen responsiveness. Castration-resistant cell line C42B showed the highest expression of most of the *IRX4* transcripts at the mRNA level. However, RNA expression does not correlate with proteomic data and complicates the precise understanding of the castration resistance and its relationship with IRX4. *IRX4* transcriptomic datasets in PCa showed a poor correlation with the proteomic data in line with recent research insights in PCa [41,61,62] due to varied discordant regulation between the transcriptional and translational regulation, post-transcriptional modifications associated with translation regulation, lack of temporal synchronization between transcription and translation and kinetic changes between protein generation and turnover in complex biological samples [63]. However, in consonance with *IRX4* RNA expression, the expression level of IRX4 protein isoforms was more prominent in the androgen-responsive cell lines than androgen-nonresponsive cell lines. Overall, the results suggest that IRX4 protein isoforms are androgen regulated. The simultaneous overexpression of *IRX4* transcripts together with *KLK3* with androgen may indicate a role of IRX4 in the early stages of PCa progression, and the antiandrogen treatment with bicalutamide and enzalutamide is not completely effective in eliminating the overexpression of *IRX4* transcripts in PCa cells. The effect of androgens and antiandrogens in the regulation of *IRX4* transcripts compared to VCaP and DuCaP cells is lower in LNCaP cells, suggestive of differential androgen regulation of *IRX4* in LNCaP cells, which needs to be further elucidated. Thus, IRX4 will be a better therapeutic target in combination with anti-androgen therapy for the treatment of PCa.

The lack of response to antiandrogen treatment of *IRX4* transcripts 6 and 7 was seen in DuCaP cells. IRX4 protein isoform 4 is encoded by transcripts 6 and 7, and these two transcripts use an alternative promoter distinct from other transcripts that are located immediately upstream of exon 3a. This may critically affect and alter the androgen regulation of IRX4 protein isoform 4 in PCa cells and can lead to therapeutic resistance to antiandrogen therapy in PCa patients. Although the function of IRX4 is tightly regulated at both transcriptional and post-transcriptional levels in human heart ventricles [64], the transcriptional regulation of IRX4 is still not completely clear in PCa. Nguyen et al. have explored the tumor suppressor role of IRX4 through the interaction with vitamin D receptors in PCa, but the isoform-specific roles are unknown [26]. Since IRX4 protein isoform 4 lacks both an N-terminus and C-terminus and predicted functional domains compared to full-length IRX4 proteins, we suspect the possibility of this isoform to act as a transcription factor. Thus, this isoform 4 may act distinctly compared to full-length isoforms by various mechanisms. Recent studies have elucidated that cancer-associated AS of transcription factors generates isoforms with altered activity, opposite transcription or antagonistic functions that severely impact tumor initiation and progression [3]. As Belluti et al. summarized, the lack of binding domains in transcription factors can show opposite functions in cancer progression [65]. For instance, the AP-2B isoform produced by AS of *AP-2* lacks a DNA binding domain and shows an inhibitor effect of the transactivation of *AP2* and leads to increased tumorigenicity, anchorage-independent growth, invasiveness and angiogenesis in melanoma [66,67]. In addition, functional domains lacking transcription factor isoforms can be mislocalized within the cancer cell and exert a dominant-negative activity as a result of the excessive expression of the non-functional isoform over functional or cytoplasmic titration of the functional isoform or regulation of full-length isoforms by non-functional isoforms directly or indirectly [65]. For example, the HELIOS-V1 isoform lacks exon 6 for the nuclear localization signal and therefore has a cytoplasmic localization in human leukemic T-cell lines and contributes to T-cell growth and survival. Further, the expression of HELIOS isoforms triggers the deregulation of various downstream target genes in T-cells compared to full-length isoforms [68]. The isoform 3 encoded from transcripts 12, 8 or 9 is expected to have the essential domains such as the transactivation domain and homeodomain, which are essential for DNA binding and act as transcription factors. However, compared to full-length IRX4 isoforms, this isoform 3 lacks an N-terminal region that may affect the structure and function in PCa, which needs to be further elucidated. Unfortunately, we could not see any protein expression in mass spectrometry data in several transcripts (10 and 11) identified in PCa cell lines that may suggest a nonsense-mediated decay of these transcripts. In view of this, although the potential importance of IRX4 isoforms towards PCa progression is still unknown, the diagnostic and therapeutic value of these isoforms cannot be ignored. Additional functional studies with isoform-specific overexpression and knockdown models are essential to prove the differential roles of IRX4 isoforms in PCa.

GWAS have identified over 160 PCa risk loci, some of which act as cis-eQTL regulatory elements which modulate the expression of nearby genes. The PCa risk SNP rs12653946 identified in the *5p15* locus was found to have a strong relationship with the *IRX4* gene (*P* = 4.91 × 10^−5^, FDR = 0.00468) [22,44]. rs10866528 has been used to tag the SNP rs12653946 and has been found to have a lower *IRX4* expression with the PCa high-risk homozygous genotype GG than with the common heterozygous AG genotype in a 50-patient sample cohort [44]. Similar to the reported results, with rs10866528 SNP we observed low levels of expression of *IRX4* transcripts in common AA genotypes and the PCa risk-associated GG genotype in a large sample cohort (483 samples). All the analyzed *IRX4* transcripts showed a similar pattern, and we were not able to observe a disparity between the analyzed transcripts. This suggests that *IRX4* transcripts equally contribute to PCa risk and one cannot ignore the individual value in the progression of PCa.

Isoform discovery is a challenging task in cancer research. Although many transcripts are identified at a cellular level, only a small proportion of transcripts encode isoforms in context-dependent manner [63]. Most of the isoforms share overlapping regions; therefore, the identification of unique isoforms is extremely difficult at a lab-based assays and computational level. The commonly used techniques, such as, PCR, RNA-seq and mass spectroscopy, have their own limitations in identifying and quantifying low abundant isoforms. We have selected a small number of PCa cell lines to quantify the expression of IRX4 isoforms by mass spectrometry, but these needed to be measured in a large sample cohort including clinical specimens. Low reproducibility, low sensitivity, high variability and high data noise of the instruments are real challenges in isoform annotation. Moreover, the limited correlation between the transcriptomic and proteomic data makes it difficult to interpret differences in isoform expression between cancer cell lines.

Analyzing isoform-specific expression in prostate tumor progression can provide clear insights to develop drugs against specific isoforms that promote tumors. Besides PCa, the diversity of the *IRX4* gene can be a hallmark for other cancers; thus, the characterization of these isoforms would augment our knowledge for the development of specific therapeutic strategies. Considering the IRX4 isoform-specific expression, our study suggests that IRX4 can have distinct roles in PCa, which suggests the clinical importance of isoform-specific therapeutic targets in improving PCa patient care.

## 5. Conclusions

In summary, identifying isoform-specific expression is a challenging but critically important task, especially in cancer progression. We identified a subset of *IRX4* transcripts in PCa whose mRNA and encoding isoforms show distinct expression profiles across a panel of androgen-responsive and nonresponsive PCa cell lines. Apart from the prominent *IRX4* 1–5 transcripts, a large contribution has been imparted by other *IRX4* transcripts to the overall *IRX4* gene expression in PCa. Some transcripts are not only lacking in regulatory and essential coding regions but also possess additional sequence features via intron retention, which suggests that the functional roles of their encoding isoforms may be distinct from the primary full-length isoforms. Given the experimental evidence associated with *IRX4* AS, our results prioritize those splicing events that can show a clear expression signal of functional importance. Thus, this highlights the importance of exploring the differential roles of potential isoforms in cancer and how cells obtain the benefit of AS in tumor progression, therapy and drug resistance. Therefore, understanding the AS of *IRX4* in PCa could provide insights into tumor development and lead to the development of new therapeutic targets.

## Figures and Tables

**Figure 1 genes-12-00615-f001:**
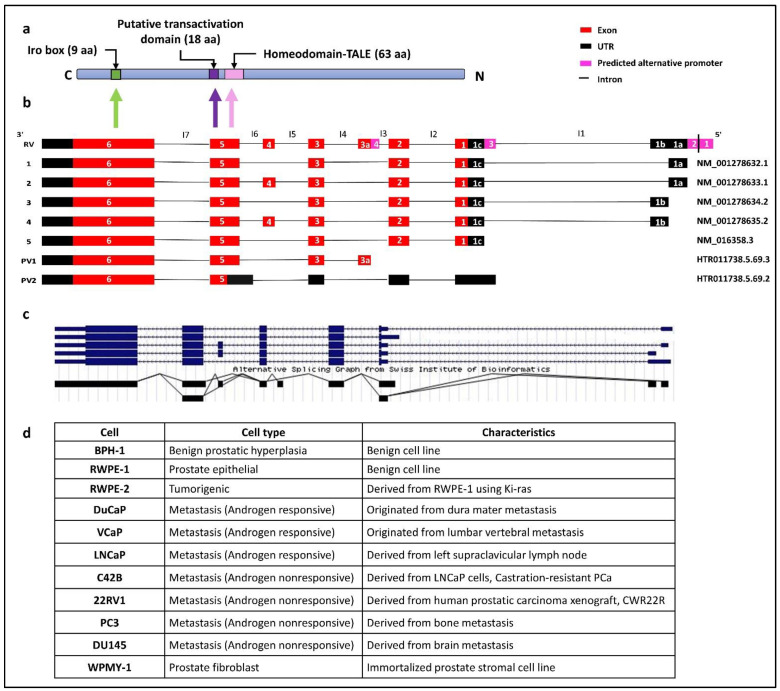
(**a**) Schematic representation of IRX4 domains. The predicted localization of the homeodomain, putative transactivation domain, irobox and the number of amino acids (aa) in different domains of the IRX4 proteins. The different domain encoding regions from the *IRX4* gene have been shown with the arrows, C-C-terminal, N-N-terminal regions. (**b**) The predicted *IRX4* transcripts according to gene databases. The five known *IRX4* transcripts: transcript 1(NM_001278632.1), transcript 2 (NM_001278633.1), transcript 3 (NM_001278634.2), transcript 4 (NM_001278635.2) and transcript 5 (NM_016358.3) and two predicted *IRX4* transcripts (PV1: HTR011738.5.69.3 and PV2: HTR011738.5.69.2) according to the Swiss Institute of Bioinformatics gene predictions and the novel putative exon (exon 3a) are presented with alignment with the Reference Variant (RV). Introns are labelled from I1 to I7. (**c**) Alternative splicing graph detailing alternative splicing (AS) events of the *IRX4* gene by the Swiss Institute of Bioinformatics, adapted from the UCSC genome browser. Lines on the plot show the exon–exon junctions. The overlapping lines denote the two types of exon–exon junction, suggestive of the presence of transcripts of *IRX4.* (**d**) The characteristics of PCa cell lines used in the study.

**Figure 2 genes-12-00615-f002:**
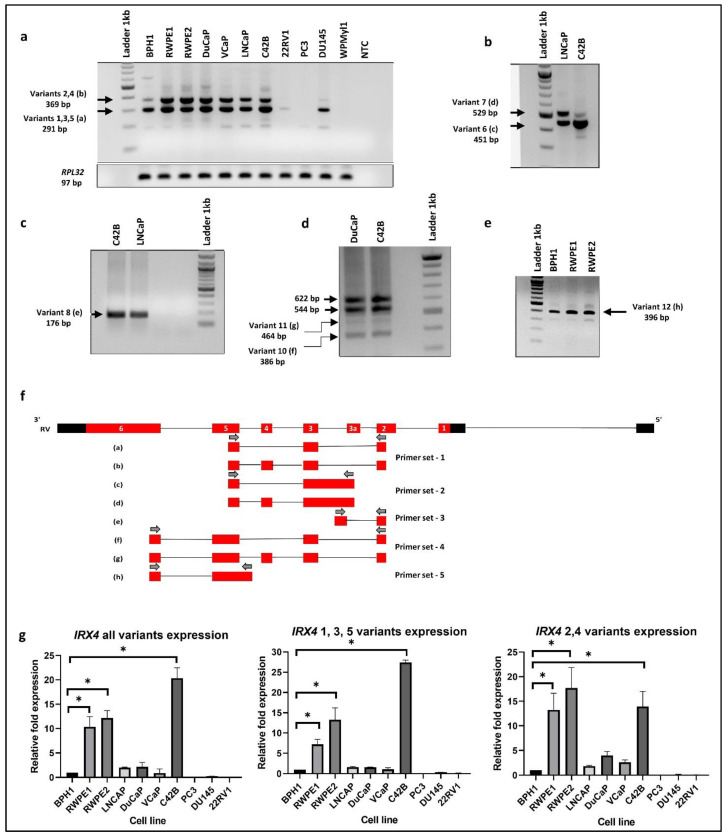
*IRX4* transcripts expression in a panel of PCa cell lines. (**a**) *IRX4* 1-5 transcripts expression (RT-PCR) from a panel of cell lines representing benign prostate (BPH1, RWPE1), PCa (RWPE2, DuCaP, VCaP, LNCaP, C42B, 22RV1, PC3, DU145) and immortalized prostate stromal cell line (WPMYl1). 1 Lane: ladder 1kb; 2–12 Lanes: cell lines; 13 Lanes: non-template control (NTC). *RPL32* was used as the endogenous housekeeping control. (**b**) *IRX4* predicted identification of transcript 6 and 7 in LNCaP and C42B cells (451 bp and 529 bp). (**c**) *IRX4* predicted identification of transcript 8 (176 bp) in C42B and LNCaP cells. (**d**) *IRX4* predicted identification of transcripts 10 and 11 in DuCaP and C42B cells (386 bp and 464 bp). (**e**) *IRX4* predicted identification of transcript 12 in BPH1, RWPE1 and RWPE2 cells (396 bp). (**f**) The sequencing BLAT results of gel bands (a) to (h) from Figure a to Figure e from the UCSC genome browser and the localization of primers re shown with arrows. The sequencing BLAT results for each transcript are presented with the alignment with the Reference Variant (RV). (**g**). The quantitative expression of *IRX4* 1-5 transcripts across a panel of prostate cell lines (BPH1, RWPE1, RWPE2, DuCaP, VCaP, LNCaP, C42B, 22RV1, PC3, DU145). *RPL32* was used as the endogenous housekeeping control. The relative fold expression was determined using the ΔΔCT method with respect to BPH1 (N = 3 biological and technical replicates, Mean ± SD, Kruskal–Wallis test with Dunn’s multiple comparisons, * *p* < 0.05).

**Figure 3 genes-12-00615-f003:**
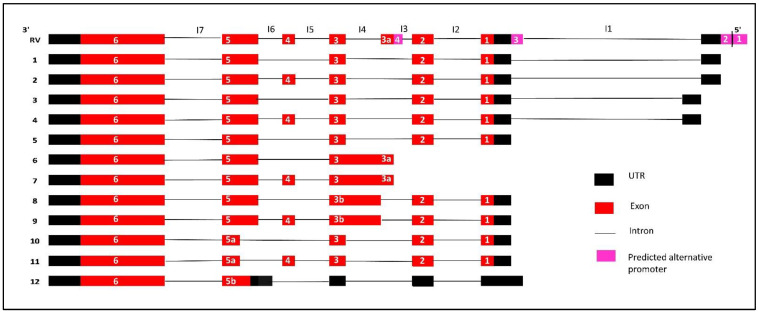
Summary of all identified *IRX4* predicted transcripts in PCa cells. All the exons and predicted UTRs are shown with red and black boxes, respectively, while introns are shown with black lines. The pink boxes indicate the location of predicted alternative promoters. Introns are labelled from I1 to I7. Each *IRX4* transcript is presented with the alignment with the Reference Variant (RV).

**Figure 4 genes-12-00615-f004:**
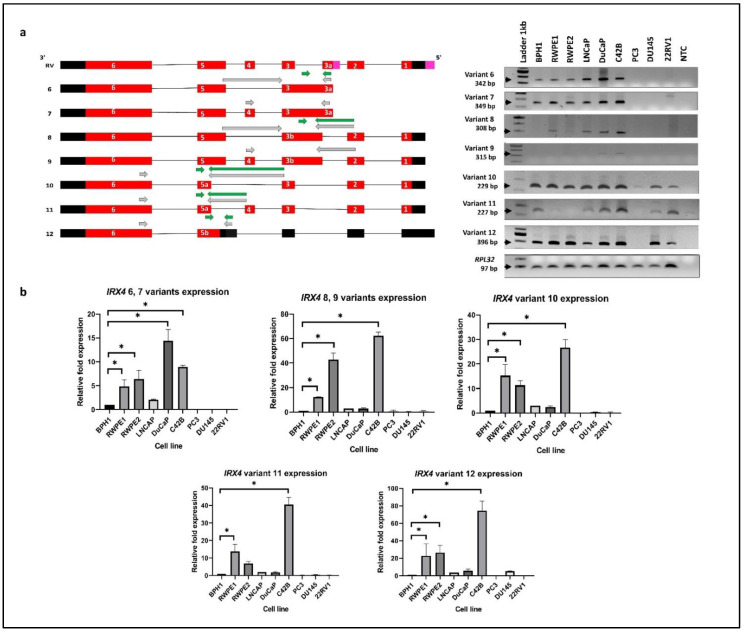
*IRX4* novel transcripts 6–12. (**a**) Qualitative and (**b**) quantitative expression across a panel of PCa cell lines. Expression analysis of each novel *IRX4* transcript with RT-PCR using transcript-specific primer sets (shown in grey arrows) and qRT-PCR using specific primer sets (shown in green arrows) in a panel of cell lines (BPH1, RWPE1, RWPE2, DuCaP, LNCaP, C42B, 22RV1, PC3, DU145). 1 Lane: ladder 1kb; 2–10 Lanes: cell lines; 11 Lanes: non-template control (NTC), *RPL32* was used as the endogenous control. The relative fold expression was determined using the ΔΔCT method with respect to BPH1 (N = 3 biological and technical replicates, Mean ± SD, Kruskal–Wallis test with Dunn’s multiple comparisons, * *p* < 0.05).

**Figure 5 genes-12-00615-f005:**
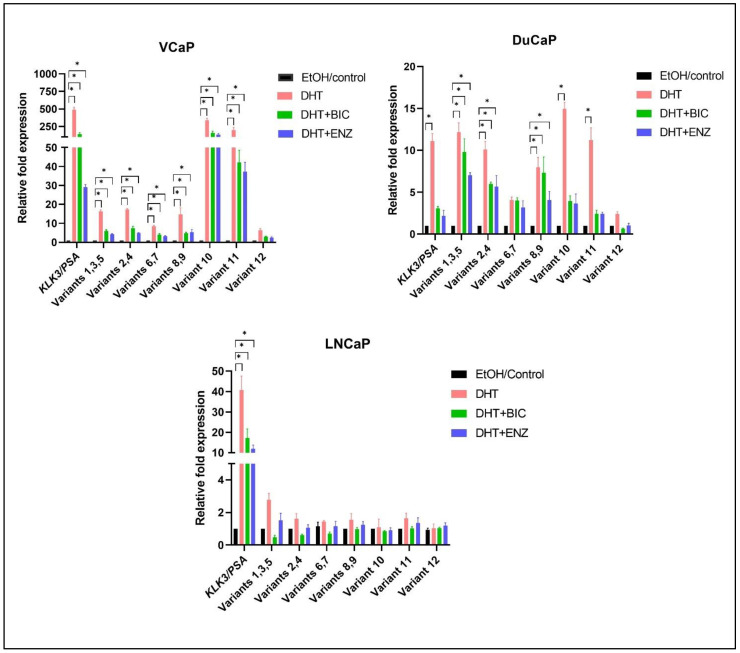
Regulation of *IRX4* transcripts expression by androgen and anti-androgen treatment in PCa cells. VCaP, DuCaP and LNCaP PCa cell lines were treated with 10 nM of androgens (DHT) or DHT + 10 µM bicalutamide (BIC) or DHT + 10 µM enzalutamide (ENZ) or EtOH/control. Relative fold expression of *IRX4* transcripts compared to EtOH/control expression was measured using the ΔΔCT method using *RPL32* as the endogenous control. (N = 3 biological and technical replicates, Kruskal–Wallis test with Dunn’s multiple comparisons, Mean ± SD, * *p* < 0.05).

**Figure 6 genes-12-00615-f006:**
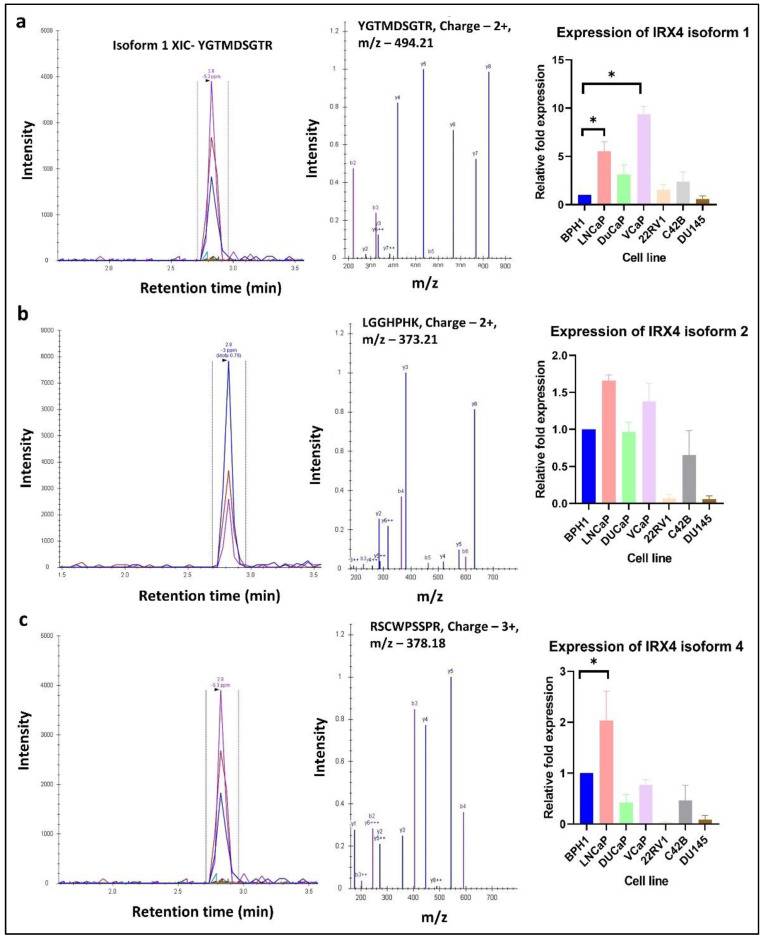
Identification and quantification of IRX4 isoform-specific peptides across a panel of prostate cell lines. Figures demonstrate the extracted ion chromatogram (XIC), MS/MS spectrum, charge and mass/charge ratio (m/z) of (**a**) IRX4 protein isoform 1, (**b**) IRX4 protein isoform 2 and (**c**) IRX4 protein isoform 4-specific peptides. The expression of each isoform-specific peptide was measured in BPH1, DuCaP, VCaP, LNCaP, C42B, 22RV1 and DU145 cell lines. The relative fold expression was measured compared to the BPH1 cell line. (N = 3 biological replicates, Mean ± SD, Kruskal–Wallis test with Dunn’s multiple comparisons, * *p* < 0.05).

**Figure 7 genes-12-00615-f007:**
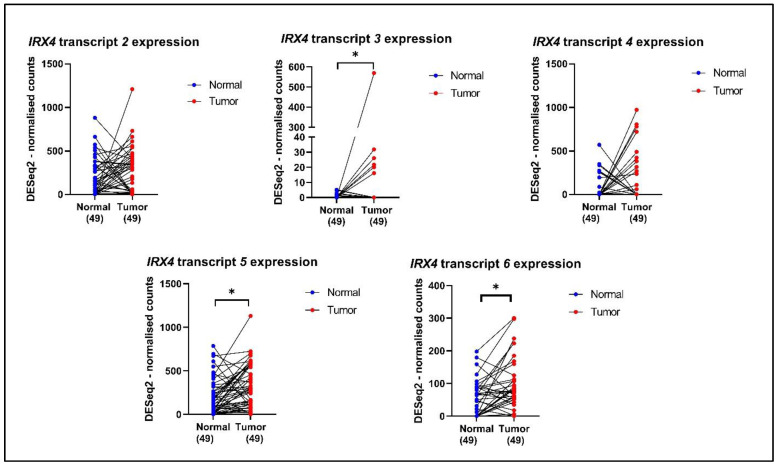
The expression of *IRX4* transcripts 2, 3, 4, 5 and 6 in PCa patient tumors and their matched normal tumor samples according to the TCGA database. A single patient is shown by a blue dot (the normal tissue) and a red dot (prostate tumor tissue), and the expression is matched with the line joining the dots. The number of patients in the normal and tumor categories is mentioned under each graph. (paired *t*-test, * *p* < 0.05).

**Figure 8 genes-12-00615-f008:**
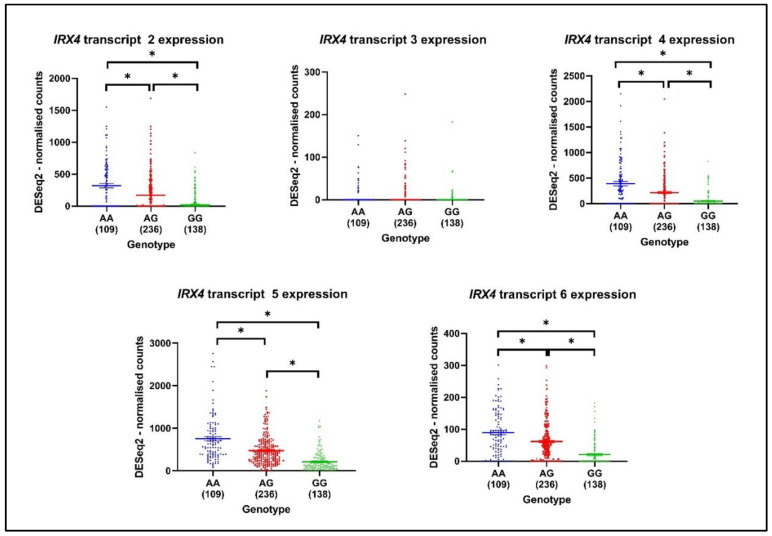
The expression of *IRX4* transcripts (mapped on DESeq2-normalized counts) by genotypes (AA, AG, GG) related to risk SNP rs10866528 in PCa. The number of patients in each genotype is mentioned under each genotype of the graph. (Mean ± SEM, Kruskal–Wallis test with Dunn’s multiple comparisons, * *p* < 0.05).

**Table 1 genes-12-00615-t001:** ExPASy tool analysis for predicted proteins from identified transcripts.

Transcript	Coding Frames	No. of Amino Acids	Predicted Protein Size	The Presence of Predicted Domains		
				Homeodomain	Transactivation Domain	Irobox
1,3,5	Exon 1–Exon 6	519	54.4 kDa (Isoform 1)	✓	✓	✓
2,4	Exon 1–Exon 6	545	57 kDa (Isoform 2)	✓	✓	✓
6,7	Exon 3a–Exon 5	114	8.7 kDa (Isoform 4)	-	-	-
	Exon 5–Exon 6	380	40 kDa (Isoform 3)	✓	✓	✓
8,9	Exon 1–Exon 3	105	11 kDa	-	-	-
	Exon 5–Exon 6	380	40 kDa (Isoform 3)	✓	✓	✓
10	Exon 1–Exon 6	204	21 kDa	✓	✓	-
11	Exon 1–Exon 6	230	23.6 kDa	✓	✓	-
12	Exon 5–Exon 6	380	40 kDa (Isoform 3)	✓	✓	✓

✓ = Presence of predicted domain.

## Data Availability

The data presented in this study are available on request from the corresponding author.

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
