# Peer review of "Identification and Characterization of Alternatively Spliced Transcript Isoforms of IRX4 in Prostate Cancer"

_genes, 2021, doi:10.3390/genes12050615_

Round 1
Reviewer 1 Report
An interesting study, but in the introduction somehow the prostate gets lost in it. The topic of the prostate should be emphasized more.
RNA purity is defined also by another ratio 260/230. Either you provide both ratios or you should write that "the RNA concentration and purity were measured using NanoDrop".
There is no information which program was used to create the primers.
An interesting choice for a housekeeping gene. Maybe researchers can explain why they choose these gene? Why the authors didn’t used other housekeeping genes which are commonly used in prostate studies: HPRT1 or GUSB?
I recommend invert the electrophoresis images, because in some pictures white stripes are not visible. Please correct to Part A of Figure 4: are very poorly visible variants of transcripts 9, 10, and 11. And in the 10 variant is not visible the marker.
Please correct lines: 129; 182; 183; 195; 198; 526.
Reviewer 2 Report
In the manuscript “Identification and Characterization of Alternatively Spliced
Transcript Isoforms of IRX4 in Prostate Cancer”, the authors combine RT-PCR/qRT-PCR, with proteomic and transcriptomic analysis of published and new datasets to characterize alternative splicing derived isoforms of the IRX4 transcription factor in Prostate Cancer cell lines and cancer patients’ samples.
They identified 7 new transcripts, and 4 new protein isoforms lacking essential functional domains of the full length IRX4 transcription factor. They also show that the different splicing variant transcripts are differentially expressed in androgen sensitive cells versus androgen-resistant cell lines. Their findings, especially regarding the different protein isoforms, suggest potential new or deleterious functions in the oncogenic context and cancer progression for these variants.
The results presented in this manuscript properly support the conclusion pointed out by the authors while also raising interesting questions about the differential involvement of the different isoforms in the oncogenic process.
Specific comments:
Figure 1:
Figure 1 right now is a summary panel of previously published information. I would incorporate in this figure a modified, schematic version of Table S2, that I believe carries critical information for the reader to move on into the analysis of Figure 2.
Figure 2:
- What is the rationale for not analyzing the expression of all the predicted isoforms (and those discovered along with the analysis) in the entire panel of cell lines? After Figure 2a, the subsequent analyses are limited to few cell lines. Are those the only ones expressing these variants upon testing? In any case, it would be informative to see the results for the entire collection of cell lines, at least as supplemental results.
- The gel in figure 2a shows additional bands, of both higher and lower MW, beyond the two pointed out. Could you comment on those? Also, the results in figure 2a don’t seem to highlight such a wide difference in the expression of the isoforms 1,3,5 in C42B versus DuCaP, VCap and LNCap cells as, instead, Figure 2g shows. The same odd correlation between 2a and 2g seems to stand also for the isoforms 2 and 4. Do you have more representative images for 2a? Or, could you comment more about this apparent discrepancy?
- Figure 2 b,c, d and e should all show a loading control as figure 2a.
- In Figure 2f it would be very helpful to have all the primer sets labeled as they are referred to in the text.
- The text refers to the new isoforms being identified and depicted in Figure 3. It would be useful to send the reader to figure 3 when naming the isoforms 6 till 12 for the first time.
Figure 5:
This figure should include the measure of the different variants, under the different treatments in at least one androgen independent cell line, to use it as a negative control.
Figure 8:
In all the experiments described till figure 8, the levels of IRX4 transcripts (all the variants) were either higher or unchanged in tumorigenic, or metastatic cell lines vs benign cell lines (Figure 2, 4, 6). This is also true for patients tumors vs matched patient normal tissue(Figure 7). However, the PCa risk genotype associates with reduced levels of the IRX4 transcripts (Fig. 8). How do you reconcile these data?
Reviewer 3 Report
This is an informative manuscript that outlines novel transcripts of IRX4 and their expression characteristics. I suggest making changes in the three following areas:
- Avoid terminology that is not commonly used, such as “risk genotype”.
- There are some grammatical issues with the paper. For example, “AS may disturb this balance with altered transcriptional regulation may end up cells with a switch between physiological to pathological transformation” is grammatically incorrect. Also, what is “physiological to pathological transformation”? Please thoroughly edit the paper to ensure proper grammar and improve clarity
- The authors described how specific IRX4 transcripts respond to androgen differently, and the differential expression levels of IRX4 transcripts. Can the authors also discuss further the implications of such differences on prostate cancer development, treatment, and therapy resistance?
Round 2
Reviewer 2 Report
The authors addressed the comments and their answers are exhaustive.
Even though the line numbers reported as references for the applied changes don't match the text, the points that the authors made in response to the reviewer's comments are acknowledged by the results and discussion sections. For this reason, I consider the article ready.